

# Comparative study of convolutional neural network architectures for gastrointestinal lesions classification

Erik O. Cuevas-Rodriguez, Carlos E. Galvan-Tejada,
Valeria Maeda-Gutiérrez, Gamaliel Moreno-Chávez,
Jorge I. Galván-Tejada, Hamurabi Gamboa-Rosales,
Huizilopoztli Luna-García, Arturo Moreno-Baez and
José María Celaya-Padilla

Unidad Académica de Ingeniería Eléctrica, Universidad Autónoma de Zacatecas, Zacatecas,
Zacatecas, México

Corresponding author
Carlos E. Galvan-Tejada,
ericgalvan@uaz.edu.mx

## ABSTRACT

The gastrointestinal (GI) tract can be affected by different diseases or lesions such as esophagitis, ulcers, hemorrhoids, and polyps, among others. Some of them can be precursors of cancer such as polyps. Endoscopy is the standard procedure for the detection of these lesions. The main drawback of this procedure is that the diagnosis depends on the expertise of the doctor. This means that some important findings may be missed. In recent years, this problem has been addressed by deep learning (DL) techniques. Endoscopic studies use digital images. The most widely used DL technique for image processing is the convolutional neural network (CNN) due to its high accuracy for modeling complex phenomena. There are different CNNs that are characterized by their architecture. In this article, four architectures are compared: AlexNet, DenseNet-201, Inception-v3, and ResNet-101. To determine which architecture best classifies GI tract lesions, a set of metrics; accuracy, precision, sensitivity, specificity, F1-score, and area under the curve (AUC) were used. These architectures were trained and tested on the HyperKvasir dataset. From this dataset, a total of 6,792 images corresponding to 10 findings were used. A transfer learning approach and a data augmentation technique were applied. The best performing architecture was DenseNet-201, whose results were: 97.11% of accuracy, 96.3% sensitivity, 99.67% specificity, and 95% AUC.

## INTRODUCTION

The human gastrointestinal (GI) tract is susceptible to different types of lesions ranging from minor annoyances to highly lethal diseases. Colorectal cancer (CRC) ranks third in cancer incidence and second in mortality (*Borgli et al., 2020*).

According to the World Health Organization (WHO), there are approximately 19.2 million new cases of cancer worldwide, of which 10% is colorectal cancer. Esophageal

cancer is the eighth most common and sixth in causes of death (*Bray et al., 2018*; *Sung et al., 2021*; *Zhao et al., 2019*).

Today, endoscopy is the diagnostic technique of choice for CRC and other lesions of the gastrointestinal tract; nonetheless, its results can sometimes be falsely negative, leading to a delay in the diagnosis of CRC (*Levin et al., 2008*). During endoscopic procedures, there are deficiencies in the detection of cancers and adenomatous polyps, as they are not easy to observe due to the different blind spots that exist in the colon (*Choi et al., 2014*; *Kaminski et al., 2010*; *Komeda et al., 2013*; *Yu et al., 2021*), with an error rate of up to 26% (*Gómez-Zuleta et al., 2021*; *Zhao et al., 2019*). In addition, approximately 50–60% of undetected lesions develop into interval cancer (*Pohl & Robertson, 2010*). Examining endoscopy videos takes quite a long time and increases the workload of expert doctors (*Öztürk & Özkaya, 2021*). Therefore, it is essential to develop computerized approaches that can assist the experts in effective diagnosis and treatment (*Owais et al., 2019*).

In Mexico, CRC represents 8.6% of cancer fatalities, just after breast cancer and a 16.35% increase is expected in 2025, according to the International Agency for Research on Cancer (IARC). The exact cause of CRC is not known, however, different risk factors increase the probability of developing it, such as changes in lifestyle and diet, *i.e.*, a higher intake of animal-based foods, excessive alcohol consumption, smoking, and a more sedentary lifestyle, leading to decreased physical activity and increased body weight (*Bray et al., 2018*; *Gómez-Zuleta et al., 2021*).

Taking into account that the diagnosis of gastrointestinal conditions is through digital images, a scenario arises in which technology and Artificial Intelligence (AI), specifically deep learning (DL) begin to show good results, the use of convolutional neural networks (CNNs) has become popular due to the ease of classifying images (*Agrawa et al., 2017*; *Chang et al., 2019*; *Hoang et al., 2018*; *Lonseko et al., 2021*).

A CNN is a neural network architecture inspired by the biological visual cortex of animals. The algorithm works with convolutional layers with shared sets of two-dimensional weights and recognizes spatial information and layer clustering to filter out comparatively more important knowledge and transmit only concentrated features (*Hiriyannaiah et al., 2020*; *Kwak & Hui, 2019*; *Song & Cai, 2021*; *Subasi, 2020*). Nowadays, there is a variety of CNN architectures (*Pacal et al., 2020*; *Alzubaidi et al., 2021*), with very different characteristics, therefore, a correct choice of architectures becomes important to perform the task of image classification of GI tract lesions. Nevertheless, these classifiers suffer from a lack of interpretability due to the fact that they are considered as "black boxes" that give good results, but without any explanation (*Gutiérrez & Tejada, 2020*). Thus, is necessary to implement a set of metrics to evaluate the performance of the architectures to comprehend the behavior of the models. Performance metrics should always be interpreted together rather than relying on a single metric (*Thambawita et al., 2020*).

However, image classification of gastrointestinal tract lesions remains a complex problem to solve because there are a limited number of databases (*Cogan, Cogan & Tamil, 2019*; *Pogorelov et al., 2017a, 2017b*), and until recently, the databases had very few images
to train the models, another factor was the quality of the images, which limited the implementation of CNNs models (*Yu et al., 2021*).

The present study proposes the implementation of four different CNNs models such as AlexNet, DenseNet-201, Inception-v3 and ResNet-101 to classify GI tract lesions and compare their performance by using a set of metrics, which are: accuracy, precision, sensitivity, specificity, F1-score, and AUC to select the architecture that best models the GI lesions classification problem.

This article is structured as follows: Section 2 describes related work of major relevance to the study. Section 3 discusses the methodology of the research work, detailing the techniques, tools, and resources used. Section 4 contains the discussion of the results obtained. Section 5 presents the conclusions of the research. Section 6 describes the future work, and finally, Section 7 reports the acknowledgments.

## RELATED WORK

In the last few years, the number of AI applications has increased exponentially. Proof of this is the remarkable advances in the field of computational image recognition, especially in the medical area, where different DL techniques have been implemented for the automatic classification of gastrointestinal lesions.

The work of *Pogorelov et al. (2017a)* presents a multiclass classification using Kvasir dataset. The dataset contains 4,000 images and eight different classes annotated and verified by expert physicians, including anatomic sites, pathologic findings, and endoscopic procedures. It uses three different approaches, the first approach uses random forest (*Macaulay et al., 2021*) and logistic model tree (*Landwehr, Hall & Frank, 2005*). The second approach uses CNNs with a rectified linear unit (ReLU) activation function and maximal clustering. The third approach is based on transfer learning, with the implementation of stochastic gradient descent (SGD) (*Hong et al., 2020*) to achieve the best performance in terms of speed and accuracy. It is worth mentioning that no data augmentation scheme was used, however, double cross-validation is implemented as a strategy to evaluate its results. The best performing approach was the logistic model tree with an accuracy of 93.70%, which combined all extracted features, resulting in a vector of 1,186 features.

*Petscharnig, Schoffmann & Lux (2017)* proposes two variations of CNN architectures with the particular feature of using an "inception" module to decrease the computational cost. The basic idea of the inception module is that the network can select at training time whether clustering, small convolution, or wider convolution is best suited to the underlying data. Kvasir is used as the dataset, a GoogLeNet-based architecture (CNN with 22 layers deep), and a data augmentation scheme (*Monshi et al., 2021*) to increase the number of images. In general, the architecture provides acceptable results even with little training data, however, the authors conclude that the model where they use 2,048 neurons in the deep layers suffers from overfitting and produces lower performance, the opposite is the case with the model where they used 1,024 neurons, obtaining an overall accuracy of 93.90%.

Likewise, *Agrawa et al. (2017)* uses Kvasir and implements a combination of pre-trained CNNs with ImageNet (*Russakovsky et al., 2015*). Employs the 16-layer configuration of VGGNet (*Simonyan & Zisserman, 2015*) as a feature extractor, using the outputs of the first fully connected layer as features for classification. An Inception-v3 network is used to extract the features and finally, a support vector machine (SVM) (*Badr et al., 2021*) is implemented for multiclass classification employing different configurations.

The hyperparameters of the SVM classifier were tuned using a five-fold cross-validation framework on the training dataset. The result was an overall accuracy of 96.10% using a combination of all features and with a data partition of 80–20% for training and testing respectively.

*Hoang et al. (2018)* combines Kvasir and Nerthus to classify 16 different classes. A ResNet with 101 layers is implemented to extract features from the original dataset, but extended with instruments. After passing through ResNet 101, the output images classified as special classes become the input to the Faster R-CNN network (*Chen et al., 2021*) that is trained to detect instruments in the images. Finally, this configuration obtained an accuracy of 99.33% and an F1-score of 94.6%, demonstrating that the use of pre-trained CNNs and a data augmentation scheme achieve good results in endoscopic image classification of the GI tract.

On the other hand, *Chang et al. (2019)* was based on learning different feature representations for multi-label images using models based on CNNs, including ResNet-34, SE-ReNeXt (*Xie et al., 2017*), and attention-Inception-v3 (*Szegedy, Vanhoucke & Shlens, 2014*). The models were trained using multi-epoch fusion and adaptive thresholding techniques with an automatic data augmentation scheme. According to the above configuration, an accuracy of 99.46%, an F1 score of 90.07%, and a Matthews correlation coefficient (MCC) of 95.20% were obtained.

The proposal of *Igarashi et al. (2020)* employs an AlexNet architecture to classify a total of 85,246 raw images obtained from Hirosaki University Hospital. The images were manually classified into 14 categories according to major pattern classification by anatomical organs. To train the model, 49,174 images of gastric cancer patients who underwent upper GI tract endoscopies were used, and 36,072 images were used to evaluate the model performance. Finally, the model obtained an overall accuracy of 96.5% and, according to the authors, the system can be used in routine endoscopy for image classification.

*Borgli et al. (2020)* presents HyperKvasir, a free-to-use database, the database has a total of 110,079 images and 374 videos of different GI tract examinations. The files are labeled images, segmented images, unlabeled images, and labeled videos with a total of 40 classes, 16 classes for the upper GI tract and 24 for the lower GI tract. To test the technical quality of the dataset, different experiments were performed with CNNs models to classify the images and the performance was measured with different metrics to give insight into the statistical qualities of the dataset. The best performing approach was the combination between ResNet-50 and DenseNet-161 both pre-trained, the average of both models was used to classify the labeled image set, which has 23 classes and 10,662 images. Both CNNs were trained with 50 epochs and a batch size of 32, and SGD was used as the optimization

method. Finally, the micro and macro averages were used to evaluate the models, and standardized classification metrics were used, resulting in an accuracy of 91% for the micro and 63.3% for the macro average.

The work of *Gómez-Zuleta et al. (2021)* presents a DL methodology for the automatic detection of polyps in colonoscopy procedures, Inception-v3, ResNet-50 and VGG-16 were the models assigned for this task. For classification, a transfer learning approach is used, and the resulting weights are used to start the new training process with colonoscopy images using the fine-tuning technique. The training scheme was a data split of 70% for training and 30% for validation, in which five different databases were combined, with a total of 23,831 and 47,013 frames with and without polyps for validation. Different metrics were implemented to measure the results, accuracy, F1-score, and ROC curve, are some of them. Finally, the Inception-v3, ResNet-50, and VGG-16 models obtained an accuracy of 81%, 77%, and 73% respectively. According to the authors, it is remarkable that these models generalize well the high variability of colonoscopy videos, moreover, this method can serve as a support for future generations of gastroenterologists.

Similarly, *Al-Adhaileh et al. (2021)* uses three networks to evaluate their potential in the classification of medical images using Kvasir as a database. First, a preprocessing is applied to remove noise from the images and improve their quality, as well as a data augmentation technique to improve the training process and a dropout technique to avoid overfitting; however, the authors mention that with this technique the training time doubled. Also, Adam is used as an optimizer to reduce loss or error, as well as a transfer learning technique and fine tuning. Finally, they are implemented to classify a total of 5,000 images with a total of five classes and a division of 80% of the database for training and 20% for validation. The models obtained an accuracy of 96.7%, 95%, and 97% for GoogLeNet, ResNet-50 and AlexNet, respectively.

Finally, *Smedsrud et al. (2021)* presents Kvasir-Capsule, a video-capsule endoscopy (VCE) dataset, which consists of 117 videos collected from endoscopic examinations with a total of 14 different classes of findings and 47,238 labeled images. A VCE is composed of a small capsule containing a wide-angle camera, light sources, batteries and other electronic components. Two CNNs, DenseNet-161 and ResNet-152 were trained to perform the technical validation of the labeled dataset, a cross-validation was implemented using categorical cross-entropy loss with and without class weighting, and weighted sampling, which balances the dataset by adding and removing images for each class. The best result was the average of both CNNs with 73.66% and 29.94% accuracy for the micro and macro average.

DL techniques are used for gastric lesion classification, as well as the diversity of approaches that exist to address classification, however, results also vary from one approach to another. CNNs show robust results and great adaptability to extract important features from gastric lesion images, moreover, with the implementation of optimization techniques the performance can be significantly improved. In this sense, the present work presents a comparative study between AlexNet, DenseNet-201, Inception-v3, and ResNet-101, selected according to the significant behavior and their reported results.
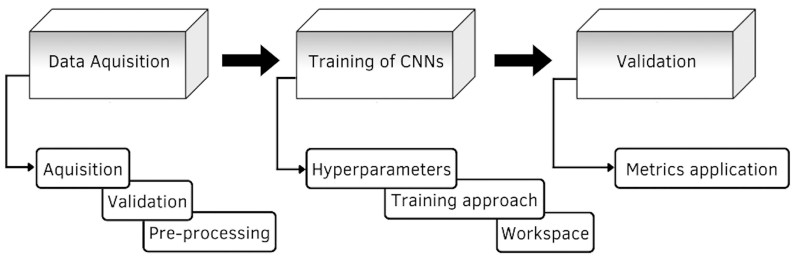

**Figure 1 Diagram of the proposed methodology.**   

# MATERIALS AND METHODS

In this section is presented the description of the research model. The research model shown in Fig. 1, consists of three stages, the first stage consists of data acquisition, here is described the construction and elements of the database, as well as its validation.
The second stage is the training of the CNNs, it describes the details of the different architectures proposed for the classification of the GI tract, it contains the configuration of the hyperparameters, such as the optimizer, the learning rate, the batch size and the number of epochs, the training approach and the image classification process are also described. The third stage is the evaluation of the models, where the performance is measured through the application of the different metrics.

## Data acquisition

One of the great challenges of AI in the medical field is the availability of data, as retrieving information from health care systems is a difficult task, as well as getting approvals from medical committees. In this regard, the HyperKvasir database aims to facilitate the development of AI in medical applications. The database is available at the following link: https://datasets.simula.no/hyper-kvasir/. HyperKvasir contains a total of 110,079 images (10,662 labeled and 99,417 unlabeled) and 374 videos of different gastrointestinal examinations.

In total, the dataset contains 10,662 images labeled with a JPEG format, of which 23 different classes are structured according to location in the GI tract and type of finding. In general, the 23 classes are separated into four main categories: anatomical locations, quality of mucosal views, pathological findings and therapeutic interventions. However, for research purposes only 10 different classes are used, selected with respect to the highest number of examples per class, as they are usually the most frequently encountered in endoscopy processes according to *Borgli et al. (2020)*.

Figure 2 shows an example of each class of the data set used, labeled as: (1) Cecum, (2) Dyed-lifted-polyps, (3) Esophagitis grade a, (4) Impacted stool, (5) Polyps, (6) Pylorus, (7) Retroflex-rectum, (8) Retroflex-stomach, (9) Ulcerative-colitis-grade-3, (10) Z-line.

Some images have a green box in the lower left corner, which is actually the topographic representation of the colon. Also, the number of images per class is not balanced due to the fact that some lesions are presented more than others, which is a challenge in the medical field. Figure 3 represents as a graph the number of images per class of the dataset. In total, 6,792 images are used to test the performance of CNNs, most of the images have a
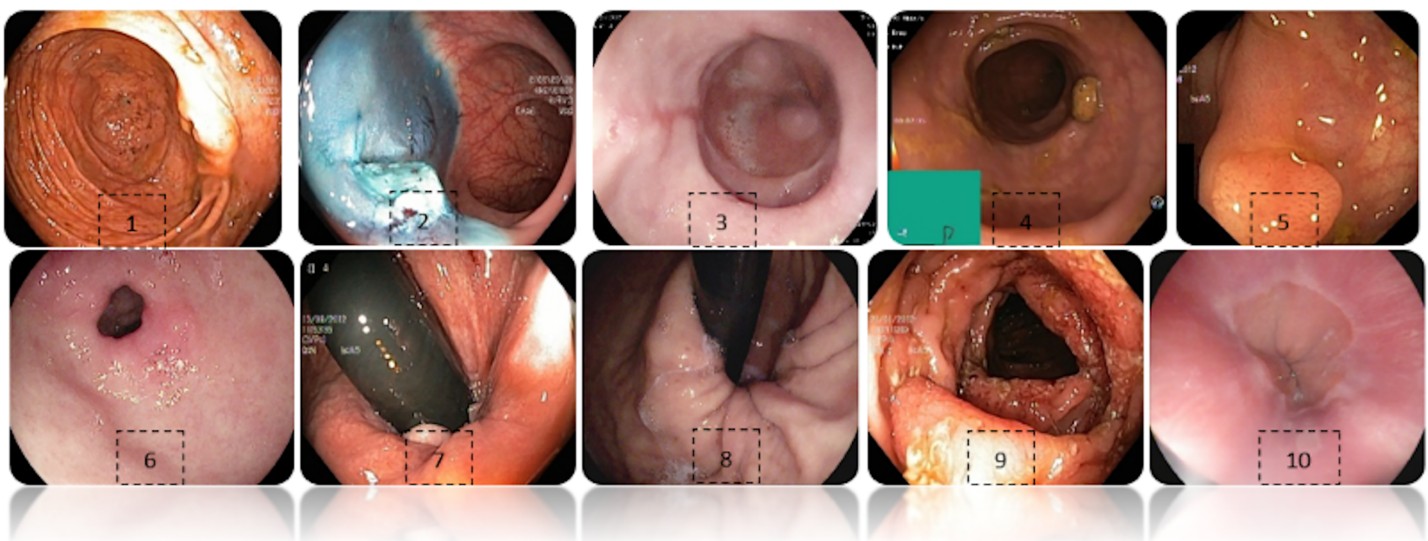

**Figure 2 Diferent classes of the dataset.** (1) Cecum, (2) Dyed-lifted-polyps, (3) Esophagitis grade a, (4) Impacted stool, (5) Polyps, (6) Pylorus, (7) Retroflex-rectum, (8) Retroflex-stomach, (9) Ulcerative-colitis-grade-3, (10) Z-line.               

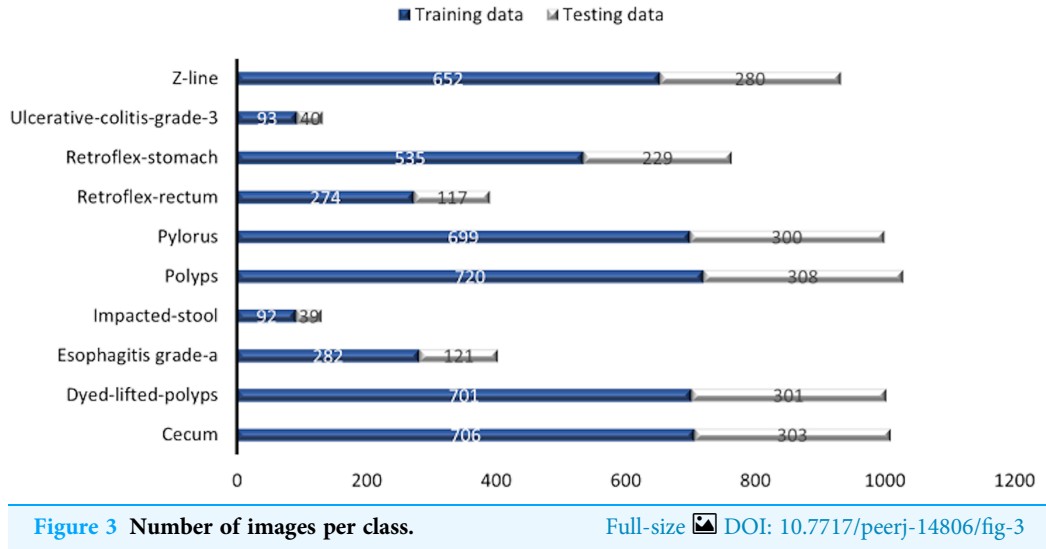

**Figure 3 Number of images per class.**               

resolution of 768 × 576 pixels. However, the image input of the CNNs has another size so a resizing is applied to adjust the image to the input size of the neural network, this is achieved by means of a bilinear interpolation (*Assad & Kiczales, 2020*). For the classification of the images, a partition of 70% was performed for training and 30% for model testing according to the review of the related work.

## Training of convolutional neural networks

According to the literature review, four models of CNNs were selected to evaluate their performance in classifying images of the GI tract, which are: AlexNet, DenseNet-201, Inception-v3 and ResNet-101. The general structure of a CNN is shown in Fig. 4.

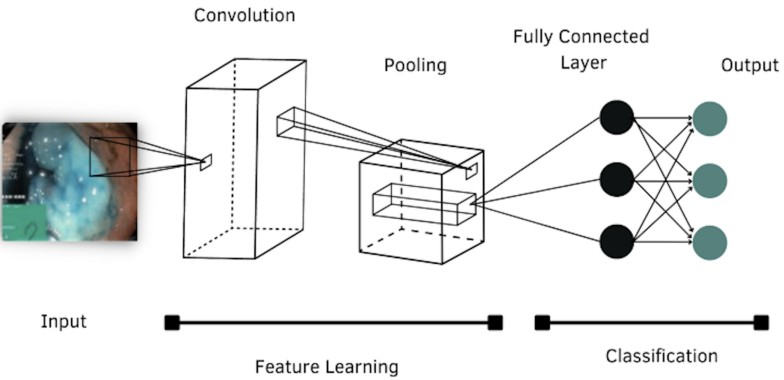

**Figure 4 Architecture of a convolutional neural network.**

### AlexNet

AlexNet was proposed by *Krizhevsky, Sutskever & Hinton (2012)*, the architecture was presented to the ImageNet Large Scale Visual Recognition (ILSVRC) and won the competition. The network has 61 million of parameters, and consists of eight layers with weights, the first five are convolutional layers and the remaining three layers are fully connected. The main feature of this network is the implementation of "dropout" as a regularization method and the use of ReLU as an activation function (*Kazemi, 2017*).

### DenseNet-201

The DenseNet model consists of many blocks and one dense block contains the convolutional layer, ReLU layer, and batch normalization. On the other hand, two dense blocks are connected to the convolutional and max-pooling layer and the last dense block is connected to the global average pooling and Softmax classifier (*Bohmrah & Kaur, 2021*). This architecture has 201 layers of deep, and works with 20 million of parameters. It needs fewer parameters than conventional CNNs because they do not need non-essential feature maps, because they are narrow and introduce new feature maps in a negligible amount (*Chauhan, Palivela & Tiwari, 2021*). To preserve the feed-forward nature each layer obtains additional inputs from all preceding layers and passes its own feature maps to all subsequent layers (*Mocsari & Stone, 2017*).

### Inception-v3

Inception-v3 is a convolutional neural network that is 48 layers deep and consists of a total of 23.9 million parameters. An Inception network is a network consisting of modules stacked on top of each other, with multiple symmetric and asymmetric building blocks, where each block has several branches of convolutions, average-pooling, max-pooling, concatenated, dropouts, and fully-connected layers to reduce the network resolution (*Szegedy, Vanhoucke & Shlens, 2014*).

### ResNet-101

ResNet models were developed by *He et al. (2016)*, they emerged as a family of deep architectures. These models obtained the first place in ILSVRC and common objects in

**Table 1 Summary of the implemented architectures.**

| Network | Depth | Size (MB) | Parameters (Millions) |
|---------|-------|-----------|----------------------|
| AlexNet | 8 | 227 | 61 |
| DenseNet-201 | 201 | 77 | 20 |
| Inception-v3 | 48 | 89 | 23.9 |
| ResNet-101 | 101 | 167 | 44.6 |

**Table 2 Hyperparameters employed.**

| Hyperparameters | Value |
|-----------------|-------|
| Learning rate | 0.001 |
| Batch size | 16 |
| Epochs | 50 |
| Optimizer | SGDM* |
| Loss function | Softmax |

**Note:**
* SGDM, Stochastic gradient descent with momentum.

Context (COCO), they differ from other architectures in terms of omission of connections and excessive use of ReLU layers (*Kazemi, 2017*). ResNet was built by several stacked residual units and developed with many different numbers of layers: 18, 34, 50, 101, 152, and 1,202. In this case, the 101-layer configuration is used, because it presents a significant increase in accuracy compared to other architectures with fewer layers. This setting works with 44.6 million of parameters.

The characteristics of each architecture are shown in Table 1.

These architectures present advanced optimization techniques that have been shown to improve training time and performance, *e.g.*, regularization methods, parameter initialization, optimizers, improved activation functions, and normalization techniques (*Johnson & Khoshgoftaar, 2019*). To compare the performance of CNNs, hyperparameters are standardized for each of the models, a good selection of these values directly affects the performance of the models, so a good choice of hyperparameters is crucial. Table 2 describes these parameters, which were selected from the review of related work and to the capability of the hardware used.

The learning rate is the speed at which an optimization function moves through the search space to converge. The batch size defines the number of data to train the models. The number of epochs refers to the backward and forward propagation correction cycle to reduce the loss. The optimizer is an algorithm used to update the network parameters at each training epoch. And finally, the loss function is used in the output layer to calculate the predicted error over the training samples, this error reveals the difference between the actual and expected output, subsequently, it is optimized through the training process of the network (*Alzubaidi et al., 2021*).

There are many reasons for using a pre-trained model. First, training models on large data sets has a high computational cost. Second, training models with many layers can be

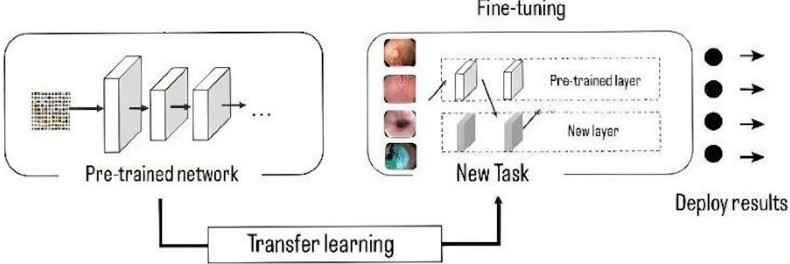

**Figure 5** **Representation of transfer learning process.**

time consuming, even taking weeks. Finally, a pre-trained model can help network generalization and accelerate convergence (*Al-Adhaileh et al., 2021*).

The models have been pre-trained in ImageNet (*Krizhevsky, Sutskever & Hinton, 2012*), a database with more than 15 million high resolution images. This approach consists of training the network using a large volume of data, where it learns the bias and weights during the training process. These weights are transferred to different networks to retrain or test a model, in this way, the new model can pre-train the weights instead of having to train from scratch, the use of these weights is done by a process called fine tuning (Fig. 5), in which the entire pre-trained network is taken and the last fully connected layer is removed. This layer is replaced by a new one, where the number of neurons is equal to the number of classes in the classification task (*Gómez-Zuleta et al., 2021*), in this case it was replaced by a fully connected layer of 10 neurons, which represents the 10 classes to be classified.

Finally, in the classification stage, a loss function is used in the output layer and calculates the predicted error over the training samples. This error reveals the difference between the actual and expected output. The Softmax function estimates the probability of belonging to a class, this function is widely used to measure CNN performance, its output is the probability $p \in \{0, 1\}$. In addition, it is commonly used as a replacement for the mean squared error function in multiclass classification problems. In the output layer, softmax activations are used to generate the output with a probability distribution (*Alzubaidi et al., 2021*).

The mathematical representation of the output class probability is the Eq. (1).

$$p_i = \frac{e^{ai}}{\sum_{k=1}^{N} e_k^a} \tag{1}$$

where $e^{ai}$ represents the unnormalized output of the previous layer, while N represents the number of neurons in the output layer.

## Evaluation of model performance

When talking about the data sets used to train CNNs, skewed data distributions arise naturally in many applications, which produces an intrinsic imbalance due to the natural frequencies of the data where the positive class occurs at a reduced frequency, including

data found in disease diagnosis (*Johnson & Khoshgoftaar, 2019*). Thus, it is necessary to properly select those metrics that best represent the performance of the models.

A metric is used to measure performance, *i.e.*, it judges the performance of the models. Currently, there is a wide variety of metrics, each of which provides specific information about a characteristic within the classifier performance. For the calculation of these metrics, it is necessary to know the number of true positives (TP), true negatives (TN), false positives (FP), and false negatives (FN).

In medical terms, positive means that the patient has abnormal lesions or has the virus, while TP means that patients with abnormal lesions are tested, and correctly labeled as abnormal lesions. Whereas FP is defined as a medical misdiagnosis of a patient with no abnormal lesions. Negative means the patient is healthy or has no abnormal lesion, and TN is the patient with no lesions and is diagnosed as normal. FN is defined as the condition where the patient with an abnormal lesion is labeled as healthy, which is a condition that causes misdiagnosis (*Wang et al., 2019*). According to the above, a more detailed analysis can be achieved based on the combination of the parameters to obtain different metrics.

The confusion matrix is a mechanism to visualize the performance of a classifier containing the four parameters defined above, the rows represent the prediction of the classifier, while the columns represent the actual value of each class (*Al-Adhaileh et al., 2021*). A more detailed analysis can be achieved by combining the parameters of the confusion matrix to obtain different metrics.

Accuracy refers to the ratio of the number of correct predictions to the total number of predictions made, and it can be calculated with Eq. (2).

$$Accuracy = \frac{TP + TN}{TP + TN + FP + FN} \tag{2}$$

Precision measures the percentage of positively labeled samples that are actually positive, and is sensitive to class imbalance because it considers the number of negative instances incorrectly labeled as positive, its mathematical representation is the Eq. (3).

$$Precision = \frac{TP}{TP + FP} \tag{3}$$

Sensitivity or recall is calculated with the Eq. (4), which allows to know the probability that a positive case is correctly classified.

$$Sensitivity = \frac{TP}{TP + FN} \tag{4}$$

Specificity is calculated with Eq. (5), which gives the probability that a negative case will be correctly classified.

$$Specificity = \frac{TN}{TN + FP} \tag{5}$$

F1-score combines accuracy and sensitivity using the weighted harmonic mean, where the coefficient $\beta$ is used to adjust the relative importance between accuracy and sensitivity, is calculated with Eq. (6).

**Table 3 Comparison of the results of each architecture.**

| Metrics | AlexNet | DenseNet-201 | Inception-v3 | ResNet-101 |
|---|---|---|---|---|
| Accuracy | 0.949 | 0.971 | 0.959 | 0.964 |
| Precision | 0.941 | 0.964 | 0.951 | 0.951 |
| Sensitivity | 0.921 | 0.963 | 0.948 | 0.949 |
| Specificity | 0.994 | 0.997 | 0.995 | 0.996 |
| F1-score | 0.924 | 0.963 | 0.949 | 0.953 |
| AUC | 0.902 | 0.949 | 0.929 | 0.945 |
| Time (min) | 99 | 1,005 | 389 | 338 |

$$F1 - score = \frac{(1 + \beta^2) \cdot recall \cdot precision}{\beta^2 \cdot recall + precision} \tag{6}$$

Finally, the area under the ROC curve is a two-dimensional graphical representation of the performance of a classifier. It is used to make comparisons between learning models and build a learning model that best models the data. In contrast to probability and metrics, the AUC exposes the classifier's overall performance.

## RESULTS AND DISCUSSION

The DL approach has been shown to enhance the performance of GI disease classification tasks significantly. This section presents and discusses the results obtained for each of the architectures. Table 3 shows the results obtained for each metric.

In general, the four architectures present a statistically acceptable overall performance, however, DenseNet-201 excels in most of the metrics, for example, it obtained 97% of accuracy, which indicates that it has a high degree of reliability in terms of the number of correctly classified predictions concerning the total number of predictions made.

In the medical field, precision is a very important parameter, since it measures the percentage of positive samples correctly classified. DenseNet-201 obtained 96.4%, while Inception-v3 and ResNet-101 obtained 95.1%, and lastly, AlexNet obtained 94.1% of precision.

Similarly, DenseNet-201 obtained 96.3% of sensitivity, ResNet-101 scored 94.9%, Inception-v3 achieved 94.8%, while the lowest performance was for AlexNet, which scored 92.1%.

In terms of specificity, all the architectures have more than 99%, which indicates how well they correctly classify negative cases.

In terms of F1 score, DenseNet-201 scored 96.3%, which indicates a good ratio between accuracy and recall, so we can say that DenseNet-201 architecture has a good balance to correctly classify positive samples. ResNet-101 achieved 95.3%, and Inception-v3 obtained 94.9%, but AlexNet scored 92.4%.

Figure 6 shows a summary of the architectures performance. It is observed how the AlexNet architecture had a lower performance than the rest of the architectures, except in specificity, where all architectures classify without problems the negative cases.

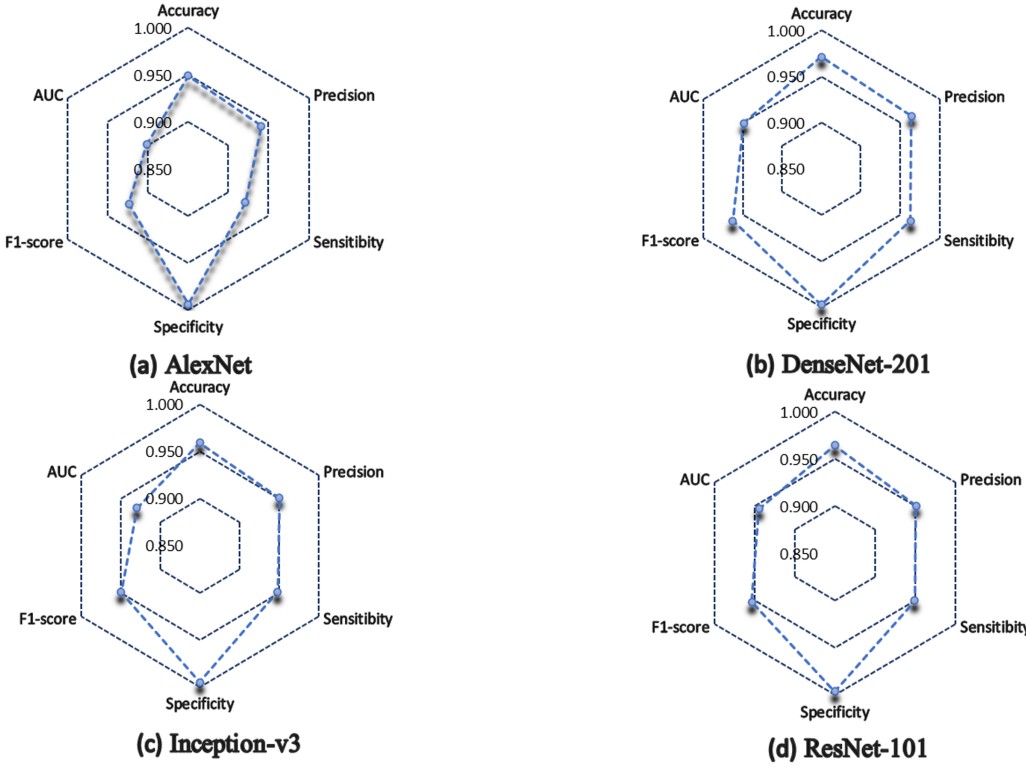

**Figure 6 Model performance summary.** (A) AlexNet model. (B) DenseNet-201 model. (C) Inception-v3 model. (D) ResNet-101 model.

Inception-v3 and ResNet-101 almost have the same performance, but in terms of AUC, ResNet-101 is superior.

DenseNet-201, the architecture with the highest number of layers, obtained the best performance in general, however, a disadvantage of using architectures with many layers is that they consume a lot of training time. DenseNet-201 was the architecture that consumed the most time, it took 1,005 min in total. Nevertheless, the experiment of the present work consisted of a single implementation of the CNNs for the classification of GI tract lesions. Therefore, training time carries less weight as a metric for model evaluation. What would be interesting would be to analyze the response time of the models when a new image is introduced.

Figure 7 shows the AlexNet confusion matrix in which a more detailed analysis of the number of instances correctly classified by architecture can be observed. It is clear how the AlexNet architecture had complications when classifying class three, which corresponds to esophagitis grade a, being able to classify only 56.19% of the instances correctly.

The confusion matrix of the DenseNet-201 model shows its ability to classify instances correctly; seven out of 10 classes were classified at 100%. The main diagonal shows the number of correctly classified samples. It can be seen in Fig. 8 that the classes with the greatest conflict were class three identified as esophagitis grade a and class 10 which corresponds to the z line. If we look at Fig. 2 it is difficult to distinguish one class from the

Peer**J**

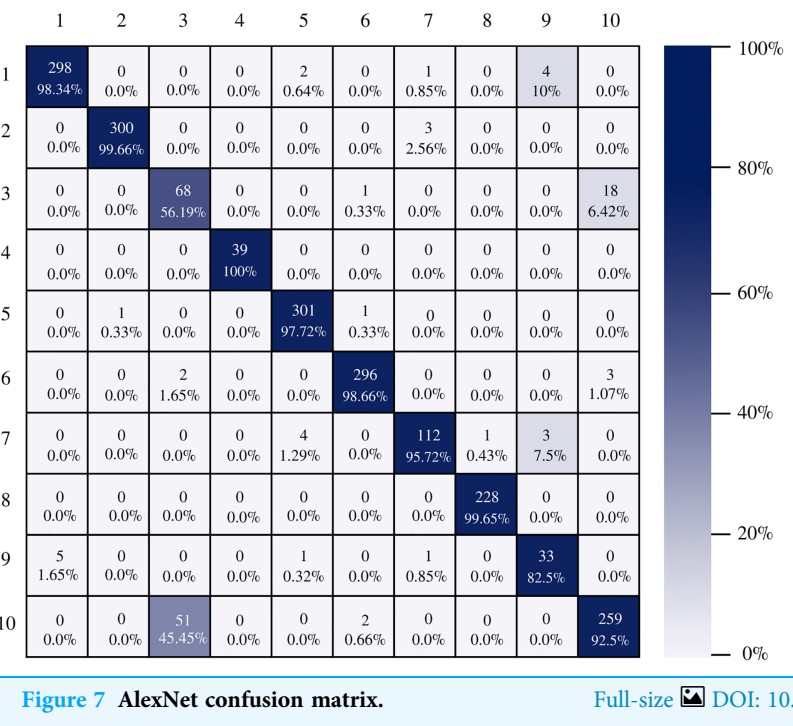
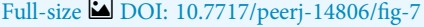

**Figure 7  AlexNet confusion matrix.**  

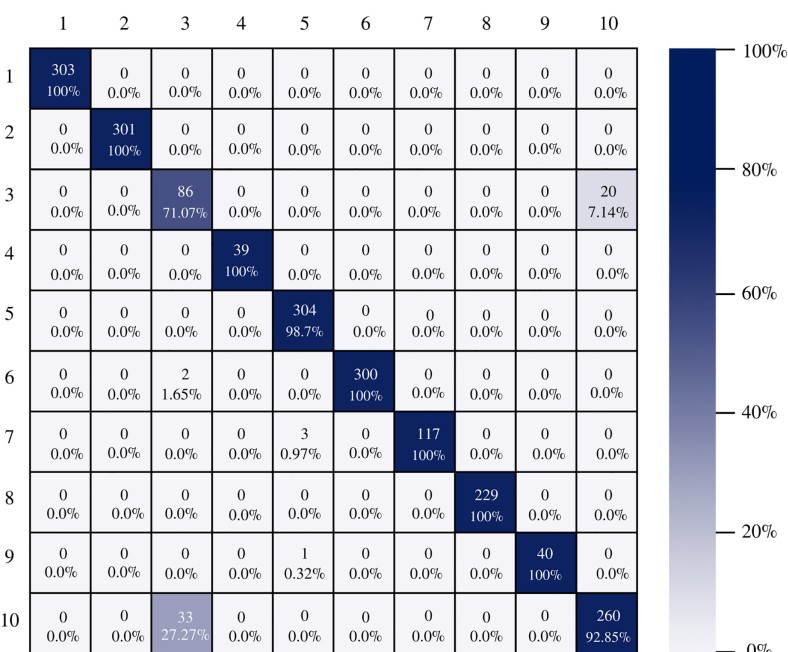

**Figure 8  DenseNet-201 confusion matrix.**

other. Added to the fact that class three contains very few samples and this implies that the model has to learn a limited number of examples, which complicates the classification.

On the other hand, Inception-v3 correctly classified all instances of only three classes. However, it obtained good percentages in the remaining classes. It can be observed in Fig. 9 that class three is still the class that generates the most conflict in the architectures.

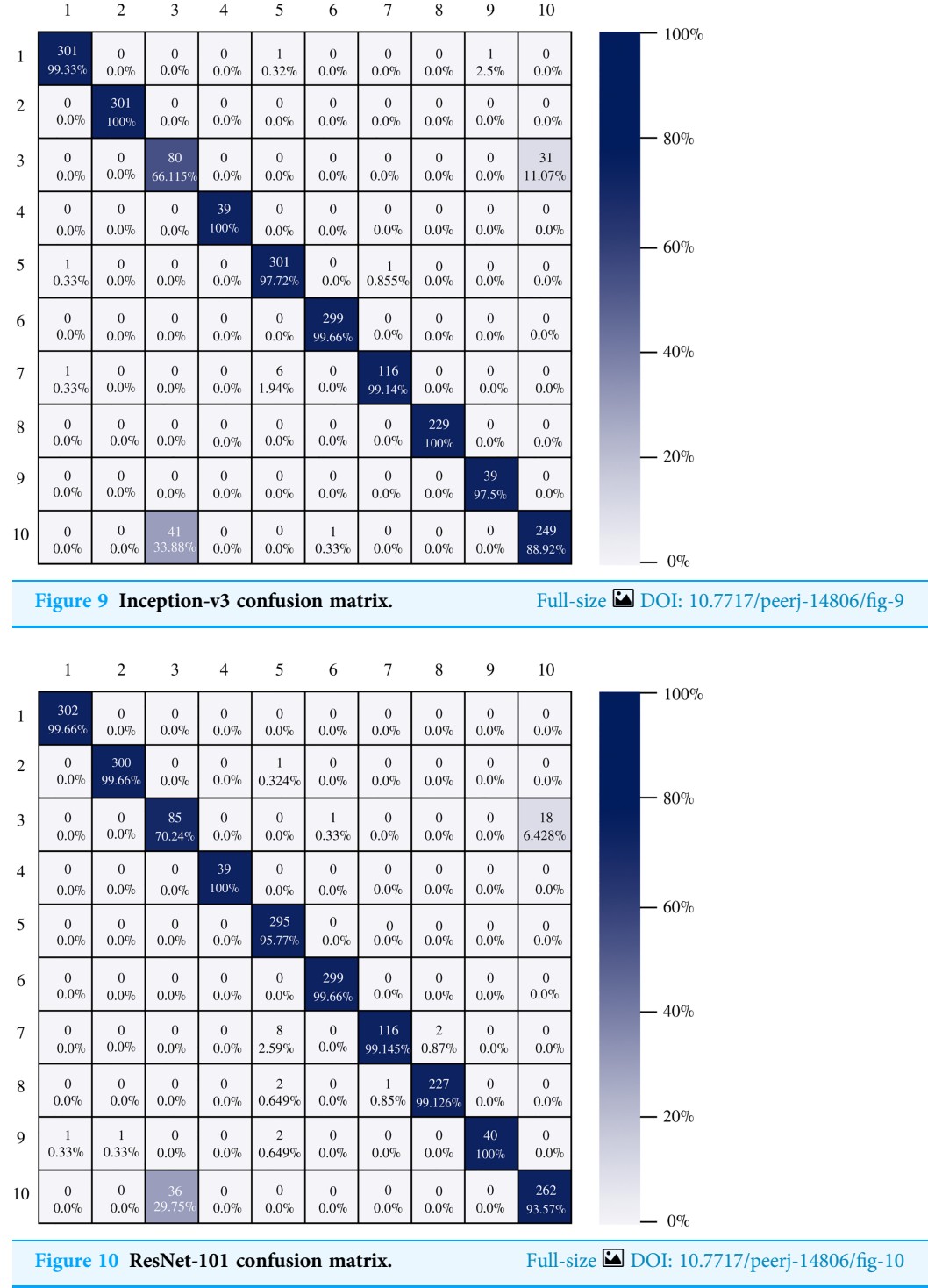

**Figure 9 Inception-v3 confusion matrix.**

**Figure 10 ResNet-101 confusion matrix.**

Figure 10 shows the confusion matrix of ResNet-101. The architecture shows good results in the classification of the instances of each class, there is a good balance between reference and prediction. However, it only correctly classified all instances of two classes, which are: impacted stool and ulcerative-colitis-grade-3.

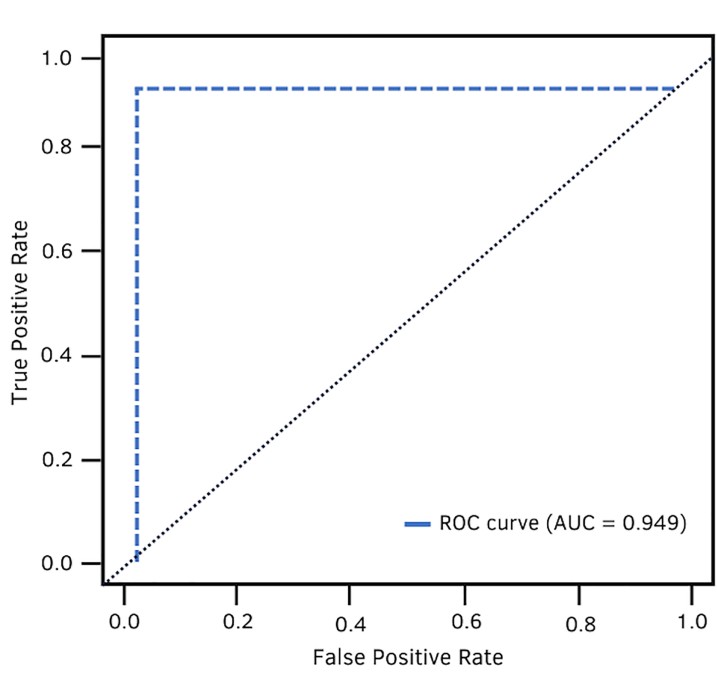

**Figure 11 DenseNet-201 multiclass ROC curve.**

In terms of AUC, DenseNet-201 achieved 94.9%. Fig. 11 shows the high classification ability of the real values. Therefore, the performance of the model is statistically reliable, as the trade-off between sensitivity and specificity is close to unity.

## CONCLUSIONS

In the present research, four architectures were selected to compare their performance in the classification of gastrointestinal lesions. These architectures were selected based on a literature review and their significant results. Each architecture has very different characteristics, from the number of layers to the type of techniques used to filter the information through the entire network. However, when evaluating their performance, the differences are minimal, so it is worth highlighting the appropriate selection of the implemented metrics and thus be able to discriminate one architecture from another.

There is no doubt that convolutional neural networks model very well the high variability that exists in images of gastrointestinal lesions. The proposed methodology demonstrates that the architectures can classify more than 90% of the samples correctly, even when working with an unbalanced database. DenseNet-201 was the best performing architecture, where seven out of 10 classes were correctly classified. This architecture excels in most metrics as we can see in Table 3 and Fig. 6. Nonetheless, it can be observed in Fig. 8 that DenseNet also had problems classifying class number three, classifying only 71.07% of the instances correctly. Affecting the overall performance of the model.

The main contribution of this work is to show the potential of four different CNNs architectures, by comparing their performance through the implementation of different metrics. The results demonstrate that the DenseNet-201 model can outstandingly

differentiate images with different lesions of the GI tract. We strongly believe that the model could be used as a computer-aided diagnostic tool, allowing more accurate diagnosis in a shorter amount of time.

One of the limitations encountered in the implementation of convolutional neural networks was the lack of images for certain classes. It was observed that the behavior of CNNs is much better when there is a large number of images to train the models.

## FUTURE WORK

A proposal for future work would be the implementation of the DenseNet-201 neural network as a support system in the endoscopy process for the identification of lesions of the GI tract.

### Funding
The authors received no funding for this work.

### Competing Interests
The authors declare that they have no competing interests.

### Author Contributions

- Erik O. Cuevas-Rodriguez conceived and designed the experiments, performed the experiments, analyzed the data, prepared figures and/or tables, and approved the final draft.
- Carlos E. Galvan-Tejada conceived and designed the experiments, performed the experiments, analyzed the data, prepared figures and/or tables, authored or reviewed drafts of the article, and approved the final draft.
- Valeria Maeda-Gutiérrez conceived and designed the experiments, performed the experiments, analyzed the data, prepared figures and/or tables, authored or reviewed drafts of the article, and approved the final draft.
- Gamaliel Moreno-Chávez conceived and designed the experiments, authored or reviewed drafts of the article, and approved the final draft.
- Jorge I. Galván-Tejada analyzed the data, prepared figures and/or tables, authored or reviewed drafts of the article, statistical interpretation of the data results, and approved the final draft.
- Hamurabi Gamboa-Rosales conceived and designed the experiments, authored or reviewed drafts of the article, statistical interpretation of the data results, and approved the final draft.
- Huizilopoztli Luna-García conceived and designed the experiments, authored or reviewed drafts of the article, statistical interpretation of the data results, and approved the final draft.
- Arturo Moreno-Baez analyzed the data, authored or reviewed drafts of the article, statistical interpretation of the data results, and approved the final draft.

- José María Celaya-Padilla performed the experiments, analyzed the data, authored or reviewed drafts of the article, and approved the final draft.

## Data Availability

The KVASIR: A Multi-Class Image Dataset for Computer Aided Gastrointestinal Disease Detection is available at: https://datasets.simula.no/kvasir/.

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
