# Peer review of "Comparative study of convolutional neural network architectures for gastrointestinal lesions classification"

_PeerJ, doi:10.7717/peerj.14806_

## Round 0.1 · original submission · Major Revisions

Thank you for the contribution. The reviewers have some comments and I had to step in because of the brevity of some revisions. Please go through the manuscript again in detail providing the answer to the raised questions.

·

Basic reporting

Thank you for this invitation. Generally, iO think this submission is acceptable, though it is pretty difficult to find one's way out of that awkward referencing system you have used. Also, I think conclusions should t the time. follow the research questions, what they do not ar

Experimental design

Appropriate

Validity of the findings

Results are sane and encouraging.

Additional comments

well presented

Reviewer 2 ·

Basic reporting

In general, I feel this warrants publication. However, I must criticize the referencing system used. It's tough to follow authors' ideas. Likewise, I think it is awkward to publish a paper on CNNs and introduce that term after a while. So, this should be changed, whatsoever.

Experimental design

no comment'

Validity of the findings

exceptional, just modify thw "conclusions" - think - what do your findings actually mean, what do you add to the science

·

Basic reporting

Generally, this is a “go-to”. Truth be told, there are some things that have to be improved

• Abbreviations are a common occurrence in research manuscripts a. However, they can also cause a lot of confusion, and make communication unclear if they are not used with caution. So, it is a general rule, all abbreviations/acronyms should be written out in full on first use (in both the abstract and the paper itself) and followed by the abbreviated form in parentheses, as in 'the American Psychological Association (APA)'.
• Usual practice with parenthetical citations (i.e., wgere both the author name and publication date are enclosed within parentheses) should include the year, When the name of the author is part of the narrative and appears outside of parentheses (like…Morin (1988) described…), after the first citation in each paragraph you need not include the year in subsequent nonparenthetical citations as long as the study cannot be confused with other studies in the article (see p. 174 in the sixth edition of the Publication Manual).
• I’m not sure I can follow “yours” difference between “increase of ” and “increase by“. Specifically, in ln 35 you say “…increase of 16.5%...”, do you mean of or by?
• Paragraphs lns 40-49 and 34-39 should change positions. I believe this could improve the narrative.
• It is a bit weird that your firs mention of Convolutional Neural Network is in the “materials and methods” ln 204.

Experimental design

extreenly good

Validity of the findings

high

---

## Round 0.2 · accepted · Accept

Congratulations and thank you in addressing the comments and suggestions properly.

·

Basic reporting

.

Experimental design

.

Validity of the findings

.

Additional comments

i think this version warrants publication. Though, it is Editor's decision

·

Basic reporting

I see a great improvement in this revised form. It's easier to follow, and key points are less vague - references are adequate, style & language as well

Experimental design

.

Validity of the findings

.

Additional comments

.